# Seipin transmembrane segments critically function in triglyceride nucleation and lipid droplet budding from the membrane

**Siyoung Kim[1], Jeeyun Chung[2,3], Henning Arlt[2,3,4], Alexander J Pak[5], Robert V Farese Jnr[2,3,6], Tobias C Walther[2,3,4,6], Gregory A Voth[7]\***

[1]Pritzker School of Molecular Engineering, University of Chicago, Chicago, United States; [2]Department of Molecular Metabolism, Harvard T H Chan School of Public Health, Boston, United States; [3]Department of Cell Biology, Harvard Medical School, Boston, United States; [4]Howard Hughes Medical Institute, Harvard T H Chan School of Public Health, Boston, United States; [5]Department of Chemical and Biological Engineering, Colorado School of Mines, Golden, United States; [6]Broad Institute of Harvard and MIT, Cambridge, United States; [7]Department of Chemistry, Chicago Center for Theoretical Chemistry, James Franck Institute, and Institute for Biophysical Dynamics, The University of Chicago, Chicago, United States

**Abstract** Lipid droplets (LDs) are organelles formed in the endoplasmic reticulum (ER) to store triacylglycerol (TG) and sterol esters. The ER protein seipin is key for LD biogenesis. Seipin forms a cage-like structure, with each seipin monomer containing a conserved hydrophobic helix and two transmembrane (TM) segments. How the different parts of seipin function in TG nucleation and LD budding is poorly understood. Here, we utilized molecular dynamics simulations of human seipin, along with cell-based experiments, to study seipin's functions in protein–lipid interactions, lipid diffusion, and LD maturation. An all-atom simulation indicates that seipin TM segment residues and hydrophobic helices residues located in the phospholipid tail region of the bilayer attract TG. Simulating larger, growing LDs with coarse-grained models, we find that the seipin TM segments form a constricted neck structure to facilitate conversion of a flat oil lens into a budding LD. Using cell experiments and simulations, we also show that conserved, positively charged residues at the end of seipin's TM segments affect LD maturation. We propose a model in which seipin TM segments critically function in TG nucleation and LD growth.

**\*For correspondence:**
gavoth@uchicago.edu

**Competing interest:** The authors declare that no competing interests exist.

## Editor's evaluation

Kim et al., investigate interactions between Seipin transmembrane domains and triacylglycerol using molecular dynamics simulations. They identify the leading steps in droplet formation and provide a physical basis for understanding the initial phases of this process, highlighting the importance of transmembrane helices in the function of seipin protein. This paper will be of interest to cell biologists and biophysicists aiming to unveil and understand how lipid droplets are formed inside cells. The topic is important given that lipid droplets are key organelles used for energy storage, and that the failure in their formation can result in various metabolic diseases.

## Introduction

The lipid droplet (LD) is a fat-storing organelle, surrounded by numerous coat proteins and a phospholipid (PL) monolayer (*Thiam et al., 2013*; *Walther et al., 2017*). LDs store excess metabolic energy

as highly reduced carbon triacylglycerol (TG) and can mobilize fatty acids for energy generation or membrane biosynthesis (*Ducharme and Bickel, 2008*; *Walther and Farese, 2012*). Due to their key role in metabolism, failure of LD biogenesis leads to metabolic diseases, such as lipodystrophy. Additionally, overwhelming the capacity of cells to form LDs is thought to be crucial for the development of diseases linked to obesity, such as fatty liver disease (*Greenberg et al., 2011*).

Current models of LD biogenesis posit that lipid droplet assembly complexes (LDACs) in the endoplasmic reticulum (ER) bilayer determine LD formation sites and facilitate LD growth (*Arlt et al., 2022*; *Chung et al., 2019*; *Prasanna et al., 2021*). LDACs, consisting of seipin and lipid droplet assembly factor 1 (LDAF1) in humans, or seipin/Fld1, Ldb16, and Ldo in yeast, efficiently catalyze the initial stages of LD formation (*Chung et al., 2019*; *Klug et al., 2021*; *Teixeira et al., 2018*; *Wang et al., 2014*). The absence of seipin, effectively removing LDAF1 as well (*Chung et al., 2019*), changes LD number and morphology dramatically, resulting in aggregated, small LDs and/or few supersized LDs (*Fei et al., 2008*; *Salo et al., 2016*; *Szymanski et al., 2007*; *Wang et al., 2016*). Therefore, investigating how seipin works is key to understanding LD biogenesis.

Human seipin is an undecamer, forming a ring structure in the ER. Each subunit contains a lumenal domain, flanked by two transmembrane (TM) segments and short cytoplasmic tails (*Arlt et al., 2022*; *Chung et al., 2019*; *Klug et al., 2021*; *Lundin et al., 2006*; *Sui et al., 2018*; *Yan et al., 2018*). The lumenal domain has a conserved hydrophobic helix (HH) thought to insert into the lumenal leaflet of the ER membrane. It was suggested that the HH of human seipin, and in particular S165 and S166, is a key tethering site for TG (*Prasanna et al., 2021*; *Zoni et al., 2021b*) and might provide a binding site of LDAF1 (*Chung et al., 2019*). Yeast seipin lacks the HH, which may explain why it is not sufficient for function in LD formation (*Arlt et al., 2022*; *Klug et al., 2021*; *Wang et al., 2014*). A recent study on yeast seipin suggests its binding partner, Ldb16, provides several serine and threonine residues in the PL tail region, thereby working as a replacement of the conserved HH of seipin (*Klug et al., 2021*).

In this study, we capitalized on new information on seipin TM segments to investigate their roles in TG nucleation and LD maturation using all-atom (AA) and coarse-grained (CG) molecular dynamics (MD) simulations. We discover a cage-like geometry of seipin TM segments facilitates a conversion of a planar oil lens into a unique ER-LD neck structure. Using cell experiments and CG simulations, we provide evidence that conserved, positively charged residues at the end of seipin's TM segments are critical for LD growth.

## Results

Seipin TM segments are thought to be critical for seipin functions (*Chung et al., 2019*). The resolved structures, however, do not include TM segments likely because of their flexibility (*Chung et al.,*

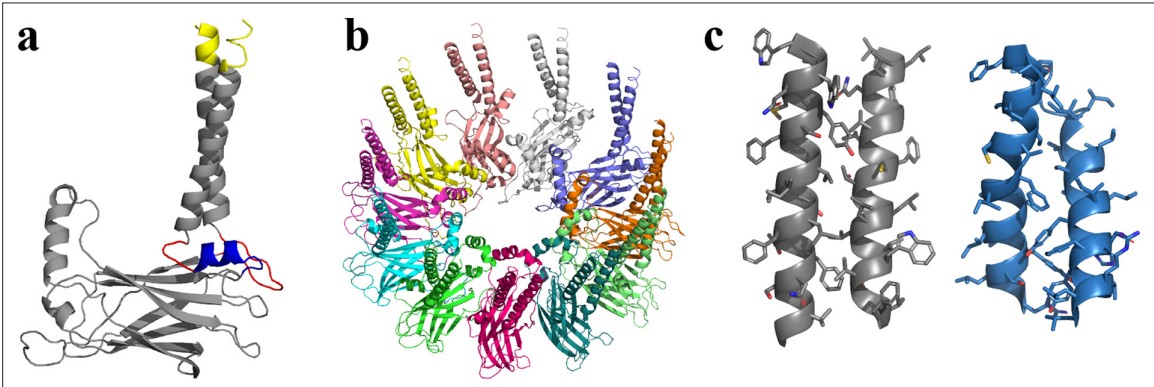

**Figure 1.** Structural model of human seipin. (**a**) Structure of a seipin subunit. The structure included in the cryoelectron microscopy data is shown in gray. Red loops were modeled with Modeller (*Fiser et al., 2000*). The blue region was predicted using the yeast structure. The helical structures were extended (yellow). (**b**) Structure of the human seipin oligomer used in the simulations. Each chain is shown with different colors. (**c**) Comparison of transmembrane segments of human seipin used in this study (gray) and yeast seipin (blue; PDB 7OXP).

The online version of this article includes the following figure supplement(s) for figure 1:

**Figure supplement 1.** Cryoelectron microscopy of human seipin.

*2019*; *Yan et al., 2018*). Therefore, we modeled seipin structure including the TM segments with the residues ranging from Arg23 to Arg265 (*Figure 1*) based on a yeast seipin structural model (*Arlt et al., 2022*; *Klug et al., 2021*) and our cryoelectron microscopy data of purified human seipin that partially resolved the TM segments (*Figure 1—figure supplement 1*; see Methods). The locations and orientations of TM segments in the cryoelectron data (*Figure 1—figure supplement 1*) were used in modeling the TM segments. However, due to the low resolution, we could not identify the residues in the TM segments. Therefore, the TM segments of the seipin model present here are subject to further validation. Next, we compared the residues of the TM segments of the resolved yeast seipin model (*Arlt et al., 2022*; *Klug et al., 2021*) and our human seipin model (*Figure 1c*). While the aromatic hydrophobic residues of the TM segments face the other subunits (outward) in the human seipin model, the nonaromatic hydrophobic residues of the TM segments are in those positions in the yeast seipin model.

How each part of seipin functions in LD biogenesis is not clearly known. To analyze the interactions between protein residues and lipids, we performed the AA simulation of human seipin in a 3-palmitoyl-2-oleoyl-D-glycero-1-phosphatidylcholine (POPC) bilayer containing 6% triolein for 3 µs. To study protein–lipid interactions, we reduced the resolution of the AA simulation by molecularly grouping each lipid or protein residue as illustrated in *Figure 2a, b*. We then calculated the normalized coordination number by molecule or the coordination number per molecule ($\langle|s|\rangle$). This quantity, thought of as the concentration-independent coordination number, indicates how much each protein residue prefers interaction with PL or TG (see Methods and *Figure 2c*). The HH exhibited a narrow spike, indicating preferential aggregation with TG (*Figure 2d*). S166 had the largest value with TG in the analysis, consistent with other computational studies using CG simulations with the Shinoda–DeVane–Klein (SDK) or MARTINI force fields (*Prasanna et al., 2021*; *Zoni et al., 2021b*). Although the modified parameters of TG have a reduced charge distribution to reproduce the interfacial tension against water (*Kim and Voth, 2021*), the TG glycerol moiety can form hydrophilic interactions with protein residues in our AA simulation (e.g., S166). In contrast, the N- and C-terminal TM segments have weaker but broader interactions with TG. Because N- and C-terminal TM segments, and HH, have helical structures, the attraction map had a weakly defined periodicity (*Figure 2d*). For instance, V163, S166, and F170 faced the membrane center, increasing the accessibility of TG. In contrast, F164 and L168 faced the lumenal interface, which prevented interactions with TG.

We further compared protein residues' attractions to TG glycerol groups or TG tail groups by normalizing the coordination number by the number of grouped atoms. While the coordination number per molecule ($\langle|s|\rangle$) indicates a propensity of each protein residue for each lipid type, PL or TG (*Figure 2c, d*), the coordination number per grouped atom ($\langle|s_A|\rangle$) provides a propensity for each group type, in this case, a TG glycerol group or TG tail group (*Figure 2—figure supplement 1*). As expected, S166 had a strong interaction with TG glycerol groups as they form a hydrophilic interaction (*Figure 2—figure supplement 1*). The alignment of the insertion depths of S166 and TG glycerol moiety likely amplified the interaction. W257, which can form a hydrophilic interaction with TG glycerol groups, had a high value as well (*Figure 2—figure supplement 1*). However, we note that those results were normalized by the number of CG atoms. If we compare the coordination number of TG glycerol groups and that of TG tail groups, TG tail groups will mostly have a higher value because there are 12 hydrophobic tail groups and 1 glycerol group for each TG molecule at the reduced resolution (*Figure 2a*). Therefore, while hydrophilic interactions at the hydrophobic phase are significant, the largest driving force of TG nucleation inside the seipin ring is provided by hydrophobic interactions of TG with seipin HH and N- and C-terminal TM segments.

To understand how seipin influences the dynamics of lipids, we computed the position-dependent diffusion coefficient relative to the center of the mass of the lumenal domain (*Figure 2—figure supplement 2*). While all lipids had comparable diffusion coefficients in the protein-free region (7.5–10.0 nm away from the seipin center), diffusion became slower near the TM segments and HH due to interactions with the protein. The decreased rate of diffusion coefficient is correlated with the contact area of protein. For instance, lumenal PLs up to 7 nm from the seipin center had the lowest diffusion coefficients because of the HH and switch region in the lumenal leaflet. In contrast, the cytosolic leaflet only contained the N- and C-terminal TM segments at the seipin boundary, leading to higher diffusion coefficients. The diffusion coefficients for TG were between those of cytosolic and lumenal PLs because TG molecules close to the lumenal leaflet can interact with the HH. In addition, strong

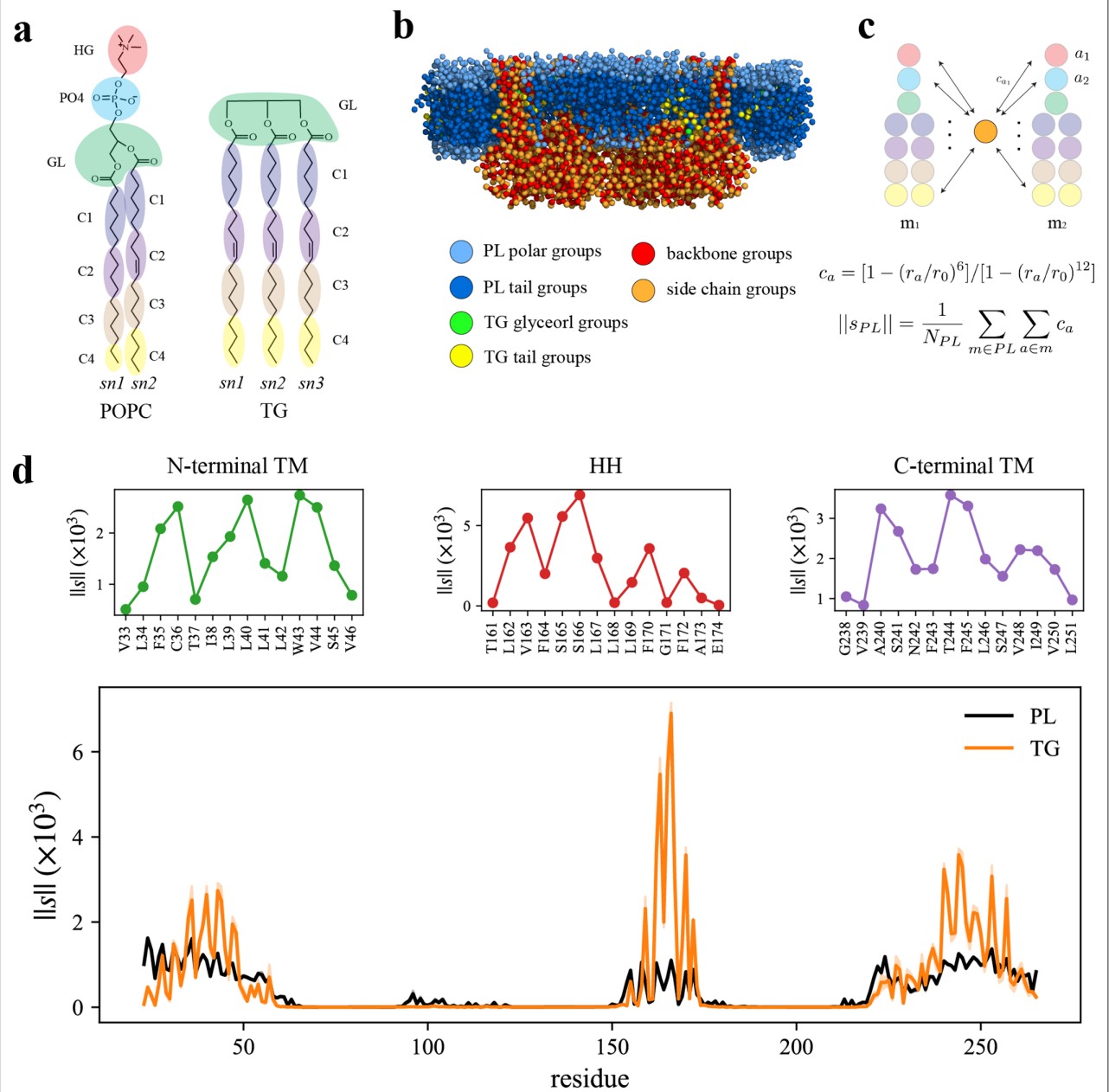

**Figure 2.** Seipin hydrophobic helix (HH) and transmembrane (TM) segments attract triacylglycerol (TG). (**a**) Molecular groupings of lipids. Each protein residue was mapped onto one side chain and one backbone molecular group. (**b**) Initial structure of the system at the reduced resolution. The snapshot was clipped in the *XZ* plane. (**c**) Illustration of the calculation of the coordination number per molecule. A side chain group was depicted with orange circle and PL groups with other colors. (**d**) Interaction plot of protein residues with phospholipid (PL) (black) and TG (orange). The shaded area represents the standard error of the results of three equal-length blocks, each containing 1-µs all-atom (AA) molecular dynamics (MD) trajectory. The residues that had high interactions with TG in the N-terminal segment, HH, and C-terminal segment were shown in separate plots in the upper panel, colored with green, red, and purple lines, respectively.

The online version of this article includes the following figure supplement(s) for figure 2:

**Figure supplement 1.** Normalized coordination number by atom.

**Figure supplement 2.** Lipid diffusion coefficients depend on location relative to the seipin oligomer.

**Figure supplement 3.** Mean squared distance of the 20 lumenal phospholipids (PLs), trapped inside the hydrophobic helix.

attractions of TG with protein residues can further reduce the rate of diffusion (*Figure 2d*). We note that the slower diffusion of the lipids that are near the protein have been discussed in previous papers (*Javanainen et al., 2017*; *Niemelä et al., 2010*). The mean squared displacement of the lumenal PLs trapped inside the seipin ring, referred to as proteinized PLs, leveled off at later simulation times due to confinement (*Figure 2—figure supplement 3*). Such confinement can increase the bending modulus of this area (*Schachter et al., 2020*), thereby working as a rigid base to ensure the direction of LD budding to the cytosolic side along with the rigid lumenal domain of seipin.

LD biogenesis is a microscopic/mesoscopic process, having its time and length scales beyond those accessible by present day AA MD simulations. For instance, during our 3-µs-long AA MD simulation we observed recruitment of TG inside the seipin complex, but not TG nucleation. To access the relevant time and length scales, we instead developed CG lipid and seipin models (*Figure 3a–c*). Linear, four-site models were used for lipids (*Grime and Madsen, 2019*; *Kim et al., 2022b*). Every four protein residues were linearly mapped to one CG atom to match the resolution with CG lipids. We also placed 24 PL molecules inside the HH ring with the orientation consistent with other PL molecules in the lumenal leaflet and considered these as a part of protein. This is based on the AA MD simulation that demonstrated PLs inside the HH ring are trapped. We constructed an elastic network model (ENM) (*Haliloglu et al., 1997*) by connecting a pair of seipin CG atoms via an effective harmonic bond with a spring constant (sc) of 0.2 kcal/mol/Å$^2$ if the distance is less than 1.5 nm (*Figure 3b*). Such a choice was made to best reproduce the fluctuations from AA MD simulation data. To understand the impact of the stiffness of harmonic bonds, we also made a model with a sc of 2 kcal/mol/Å$^2$. To achieve the known stability of seipin in a bilayer membrane, nonbonded protein–lipid interactions were based on the lipid–lipid interactions. Attraction scaling factors that change the force and potential depth linearly are shown in *Figure 3c* (*Kim et al., 2022b*). Although it is not obvious how to quantitatively incorporate the AA MD simulation data into phenomenological CG models, higher scaling factors between TG-HH and TG-TM can be qualitatively justified by the high attractions of those pairs indicated in the analysis of the AA MD simulation (*Figure 2d*).

Seipin is thought to catalyze TG nucleation, thereby decreasing the critical concentration (*Chung et al., 2019*). To test this hypothesis, we performed the CG simulations of spherical bilayers with a diameter of 40 nm. The radius of the spherical bilayer is comparable that of the actual ER tubule (*Georgiades et al., 2017*). The initial structures had evenly distributed 2 mol% TG molecules. Because the TG concentration is below the critical concentration for its phase transition (*Hamilton and Small, 1981*; *Khandelia et al., 2010*; *Zoni et al., 2021a*), TG nucleation did not occur in the lipid system (*Figure 3d*). In contrast, the system that includes the seipin complex showed a nucleated TG lens inside the complex due to the attractions between TG-TM and TG-HH (*Figure 3d*). As a control simulation, we included a single seipin subunit in the lipid system and carried out the CG MD simulation. TG nucleation did not happen in the system (*Figure 3d*), indicating that a single subunit falls well short of TG nucleation. Therefore, high protein density at the seipin site provides for collective and cooperative interactions with TG, catalyzing TG nucleation. To support our findings of seipin's role in facilitating TG nucleation, we also performed MARTINI CG model simulations in a bilayer, containing 2% TG (*Figure 3e*). While TG was dissolved in the reference lipid system without seipin, TG formed its distinct phase inside the seipin complex. Such seipin-driven TG nucleation was observed in the recent other computational studies using MARTINI or SDK CG force fields (*Klug et al., 2021*; *Prasanna et al., 2021*; *Zoni et al., 2021b*). Given that these studies used different approaches to model the TM segments but came to the same conclusion, it is expected that the lumenal domain of seipin, especially the HH, is critical in TG nucleation, while the locations and orientations of the TM segments contribute less to TG nucleation. Interestingly, yeast seipin that lacks the HH could not facilitate TG nucleation in the simulations (*Klug et al., 2021*), consistent with the experimental observations in which yeast seipin alone is not functional (*Wang et al., 2014*).

Exploiting the computationally highly efficient nature of our low-resolution CG models, we also investigated LD maturation. In particular, to study the impact of the cage-like structure of the seipin oligomer and their TM segments in LD biogenesis, we simulated various geometries of seipin in the spherical bilayers containing 6% TG (*Figure 4*). In the first model, we removed the TM segments, and the resulting model only contained the lumenal domain. In the second model, we removed six continuous subunits from the seipin oligomer, and the resulting model contained five subunits. As a reference, we also carried out simulations of the pure lipid system and the 11mer-containing system.

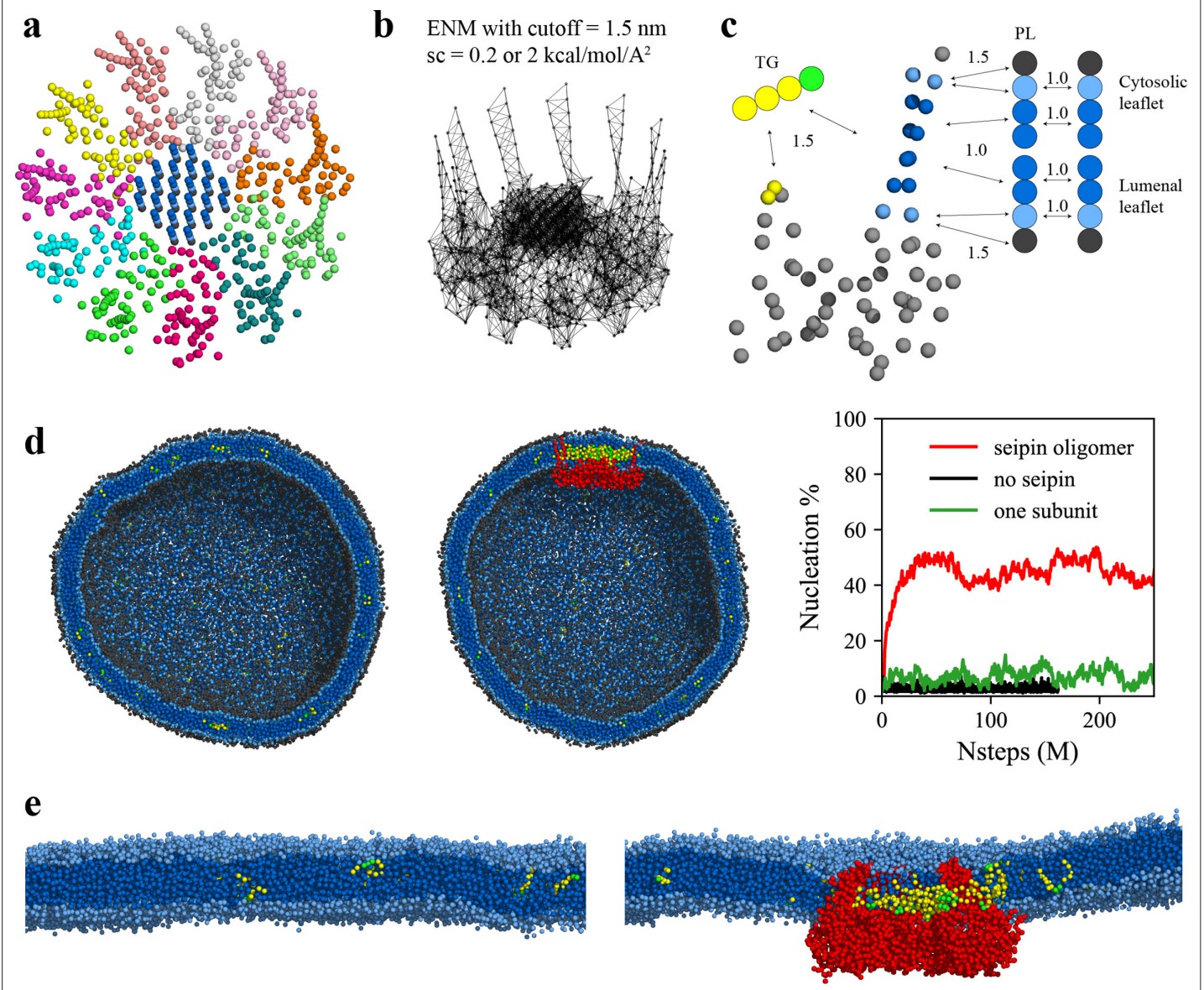

**Figure 3.** Seipin lowers the critical concentration of triacylglycerol (TG) nucleation. (**a**) Coarse-grained (CG) model of human seipin oligomer. The CG atoms inside the hydrophobic helix ring represent phospholipid (PL) atoms (proteinized PLs). (**b**) Elastic network model (ENM) with a spring constant of 0.2 or 2 kcal/mol/Å². (**c**) Scaling factors of attraction parameters between seipin–PL and seipin–TG interactions. PL head, interfacial, and tail groups are shown with black, light blue, and dark blue, respectively. TG glycerol and tail groups are shown with green and yellow, respectively. Seipin atoms that attract PL tails are shown with dark blue, and those that attract PL interfacial atoms are shown with sky blue. Two seipin atoms in the hydrophobic helix (HH), shown with yellow, and four central seipin atoms in each transmembrane (TM) segment, shown with dark blue, attract TG atoms. (**d**) CG molecular dynamics (MD) simulations of bilayers containing 2% TG with a diameter of 40 nm were carried out. The clipped snapshots of the last frames of the pure lipid (left) and seipin-containing systems (center) are shown. The ENM model with a spring constant of 0.2 kcal/mol/Å² was used. PL head, interfacial, and tail atoms are shown with black, light blue, and dark blue, respectively. TG glycerol and tail atoms are shown with green and yellow, respectively. Seipin oligomer is indicated with red. The nucleation percentages of three systems with simulation steps are shown in right. (**e**) MARTINI CG model simulations of TG nucleation. TG does not nucleate in the bilayer without seipin (left), while it does in the bilayer with seipin, forming its distinct phase inside the seipin complex (right). Both bilayers have 2% TG, lower than the critical concentration.

Because the TG concentration was above the critical concentration, TG nucleation occurred in those systems even in the system without seipin. However, the resulting morphologies of oil lenses of those systems were different as shown in the final snapshots and characterized by anisotropy. First, in the lipid system, a nucleated TG lens was flat and had high anisotropy to minimize the membrane deformation penalty (*Kim et al., 2022b*). In contrast, in the 11-subunit model, a nucleated TG lens was

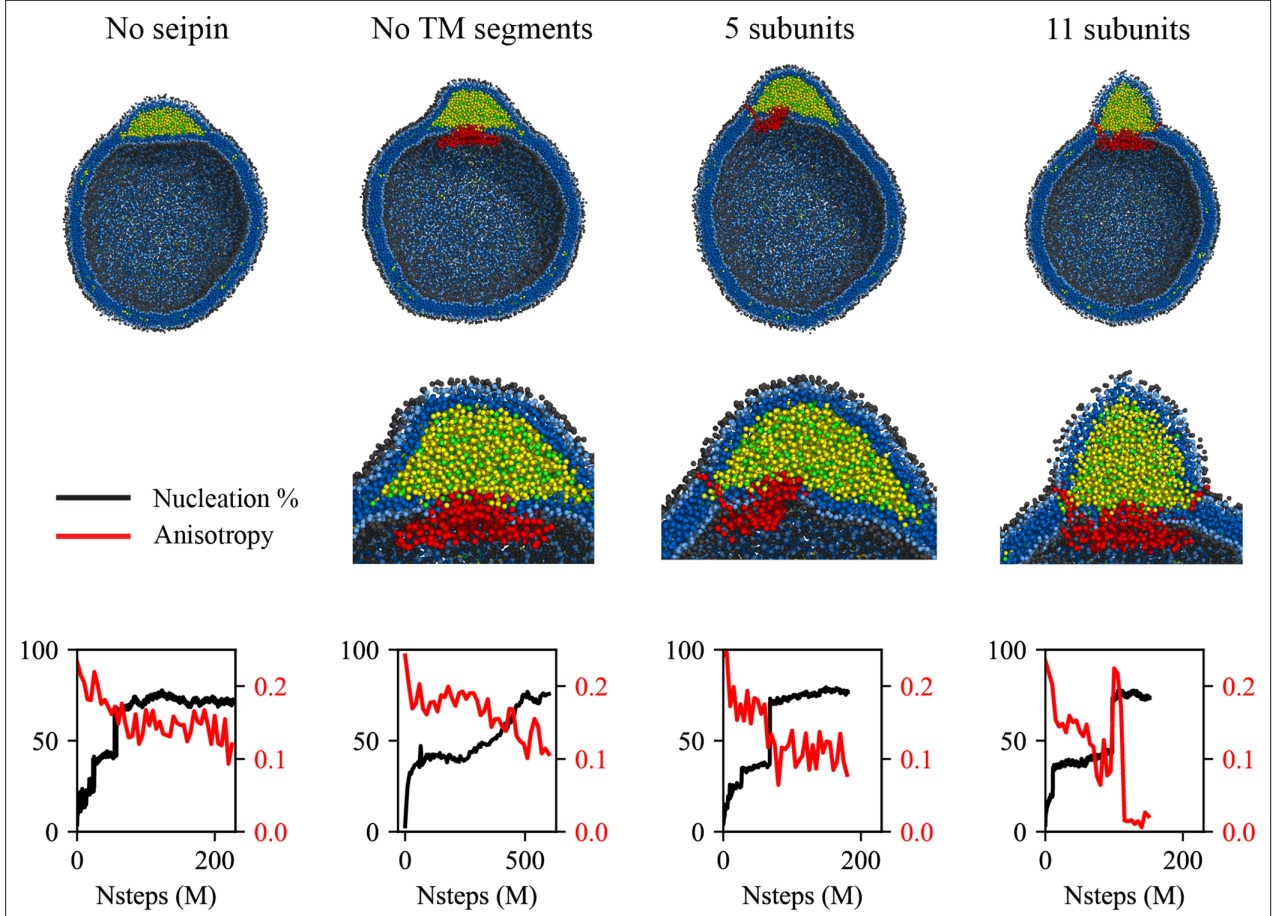

**Figure 4.** Cage-like geometry and neck formed by seipin transmembrane (TM) segments are key to modulating the morphology of a forming oil lens. The first row shows the clipped snapshots of the equilibrated frames, and the second row shows the closeup view of the seipin. The third row shows the nucleation percentage (black) and anisotropy (red). 'CG simulations of spherical bilayers containing 6% TG with a diameter of 40 nm were carried out'. The elastic network model (ENM) model for seipin with a spring constant of 0.2 kcal/mol/Å² was used.

The online version of this article includes the following figure supplement(s) for figure 4:

**Figure supplement 1.** The diameter of an oil lens is shown with simulation steps.

**Figure supplement 2.** Coarse-grained (CG) simulation with a spring constant of 2 kcal/mol/Å².

located on top of the seipin lumenal domain, surrounded by seipin TM segments. This resulted in a significant change in the shape of the oil lens from high anisotropy in the lipid system, minimizing the membrane deformation penalty, to low anisotropy in the seipin-containing systems. Given the nucleation percentages were comparable in those simulations, a change in anisotropy can be attributed to the seipin TM, not to the amount of nucleated TG molecules. Importantly, the seipin TM segments constrained the XY area where TG can be in the bilayer, pushing excessive TG molecules to the budding LD. This results in the formation of the ER-LD neck structure, consistent with an experimental tomogram (*Salo et al., 2019*). An equilibrated diameter of the ER-LD structure (*Figure 4—figure supplement 1*), approximated by a diameter of the circle formed by the end residues of N- and C-terminal TM segments, also agreed well with the experimentally measured data, which is in the range of 13–17 nm (*Salo et al., 2019*). We note the TM segments tilted away from the oligomeric center during LD growth; therefore, the diameter of the seipin ring increased with simulation time. We also simulated the 11-subunit model with a spring constant of 2.0 kcal/mol/Å² (*Figure 4—figure supplement 2*). The higher spring constant reduced the diameter of the ER-LD neck structure. However, the nucleation percentage and morphology of the formed oil lens had little difference with the previous system that contained the ENM with a spring constant of 0.2 kcal/mol/Å².

The 5-subunit model can be considered a mixture of the lipid system and 11-subunit model because one end is occupied with seipin subunits while the other end is exposed to lipids. The resulting morphology of an oil lens from the CG MD simulations was also between those results. The TG oil lens was elongated to the region where there was no seipin subunit; however, it was constrained in the region of seipin subunits, especially by their TM segments. The equilibrated anisotropy was also between the lipid and 11-subunits systems. Finally, we simulated the seipin model that only contained the lumenal domain. Such a complex does not form a mobile focus in cells likely because it fails to form an oligomer, it is not stable in bilayers, or it is degraded (*Chung et al., 2019*). However, simulating this system can provide further valuable insight on the roles of the TM segments. The resulting oil lens showed little difference from the lipid system without seipin. The anisotropy was high, and the formation of the ER-LD neck structure was abolished.

It is worth noting that the mechanisms of oil growth can be predicted from analysis of the nucleation percentage and anisotropy. A sharp increase in the nucleation percentage indicates oil coalescence, as shown in ~70 M MD time steps in the 5-subunit system and ~100 M MD time steps in the 11-subunit system (*Figure 4*). In contrast, the nucleation percentage grew slowly in the seipin model without the TM segments from ~200 M MD time steps, indicating Ostwald ripening. Finally, a sudden increase followed by a sharp decrease in anisotropy in the 11-subunit system implies a slow coalescence. When any two TG molecules in distinct oil lenses are within 2 nm, those oil lenses are considered one oil lens, as shown by a step increase in the nucleation percentage at 100 M MD time steps (*Figure 4*). Such a snapshot resembles two humps, increasing anisotropy. After ~110 M MD time steps, the oil lenses are fully merged into one spherical lens, reducing anisotropy to zero. Controlled coalescence at the seipin site will be discussed later.

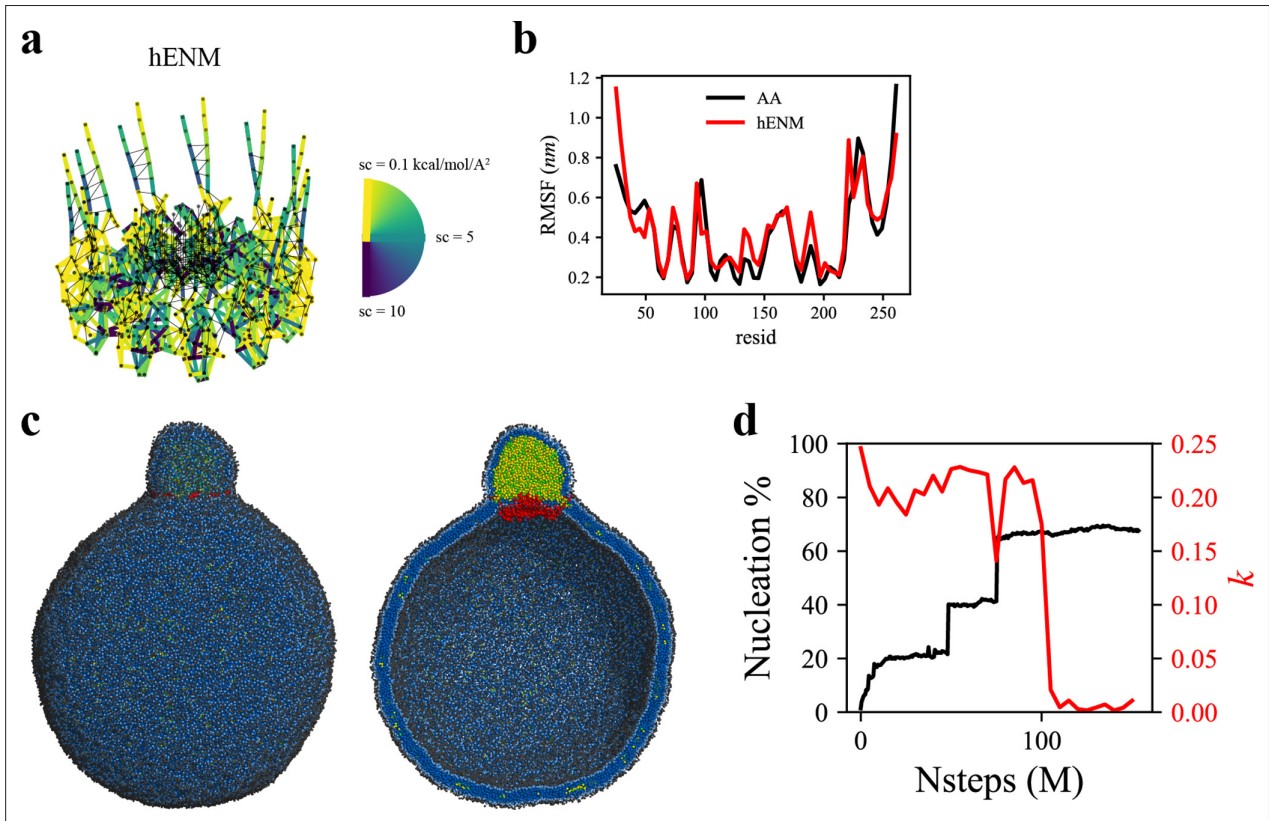

**Figure 5.** Coarse-grained (CG)-molecular dynamics (MD) shows lipid droplet (LD) growth in a large bilayer. 'CG MD simulations of spherical bilayers containing 6% triacylglycerol (TG) with a diameter of 60 nm were carried out'. (**a**) Heterogeneous ENM (hENM) model of human seipin was constructed. Pairs of atoms in a subunit were connected with harmonic springs with their spring constants (sc) represented by their color. Additional harmonic springs (black lines) with a spring constant of 0.1 kcal/mol/Å² were added between pairs of atoms that were not included in the hENM with a distance cutoff of 11 Å to ensure connections between subunits. (**b**) RMSF of a seipin subunit was compared between the all-atom (AA) trajectory and hENM in a bilayer. (**c**) Exterior and interior view of the last frame of the CG simulation. (**d**) The nucleation percentage (black line) and anisotropy (red line) are shown.

To investigate the more advanced biogenesis steps, we simulated a larger spherical bilayer with a diameter of 60 nm and 6% TG. We also constructed a hENM model (*Lyman et al., 2008*) using the fluctuations obtained from the AA simulation of seipin in the bilayer membrane (*Figure 5a*). The calculated root-mean-square fluctuations (RMSF) from the AA and CG simulations agreed well in bilayers (*Figure 5b*). Consistent with the previous results, the seipin TM segments defined the oil boundary, facilitating the transport of TG into the LD (*Figure 5c*). The equilibrated anisotropy was close to zero, indicating a spherical shape of the forming oil lens (*Figure 5d*). Collectively, our tests demonstrate that the ring geometry of seipin TM segments is key to defining the boundary of the forming oil lens and creating the unique ER-LD structure.

The coevolutionary sequence analysis (*Hopf et al., 2019*) of human seipin indicated that seipin's lumenal domain and two TM segments are evolutionarily conserved (*Cartwright and Goodman, 2012*; *Figure 6—figure supplement 1*). In contrast, the N- and C-terminal tail regions, exposed to the cytosol, are not conserved (*Figure 6—figure supplement 1*). To test whether the nonconserved, cytosolic tails are important for LD formation, we experimentally constructed seipin-ΔTERM, which lacks the N-terminal region (1–22 amino acids) and C-terminal region (268–398 amino acids). First, we confirmed that seipin-ΔTERM has the same membrane topology with wildtype protein (*Figure 6a*) and assembles discrete seipin foci in the ER. We next tested whether seipin-ΔTERM is functional by examining its ability to rescue the LD phenotypes in seipin knockout (KO) SUM159 cell line we previously reported (*Wang et al., 2016*). Consistent with the previous experiments (*Chung et al., 2019*; *Fei et al., 2008*; *Salo et al., 2016*; *Szymanski et al., 2007*; *Wang et al., 2016*), seipin KO cells had massive accumulation of small nascent LDs at the early time (~1 hr) of LD formation after oleate treatment (*Figure 6b, c*). Seipin-ΔTERM rescued the defective LD phenotype of seipin KO cells (*Figure 6b, c*), indicating that the cytosolic nonconserved N-/C-terminal regions of human seipin are dispensable for the seipin function. This is comparable with previous experiments that demonstrated the removal of N- or C-terminal region of fly seipin in *Drosophila* S2 cells did not change the rescue efficiency of the seipin deletion phenotype (*Wang et al., 2016*).

We also investigated if the positively charged residues located at the beginning of the N-terminal TM segment (R23, R24, and R26) and the end of the C-terminal TM segment (R265, H266, and R267) are essential for the seipin function. Those charged residues located at the borders of the TM segments are conserved (*Figure 6—figure supplement 1*), and in our CG trajectories, they were exposed to the outer leaflet of the membrane (*Figures 4 and 5c*). To test the importance of those residues, we experimentally made the two mutant seipin constructs, seipin-FL-tipA and seipin-ΔTERM-tipA, in which their arginine and histidine residues at the ends of seipin's TM segments were mutated to alanine. Despite the absence of charged residues, those seipin constructs are correctly inserted into the ER bilayer and stably formed foci as same as wildtype seipin (*Figure 6a*). However, seipin tipA mutants were not fully functional and resulted in an intermediate LD phenotype with more and irregular shaped LDs compared to wild-type cells (*Figure 6b, c*).

We further hypothesized that the reduced attraction between the borders of seipin's TM segments and the membrane interface caused defective LD maturation in the cells that contained the mutant constructs. To investigate this possibility, we carried out CG simulations with variable attraction scaling factors (*r*) between the residues at the membrane interface and PL interfacial and head groups (*Figure 6d*). When *r* is small, it could be thought of as the mutant construct that does not have the charged residues at the ends of TM segments (seipin-ΔTERM-tipA). The resulting CG simulations demonstrated that those residues no longer interacted with the interface of the cytosolic leaflet (*Figure 6d*). Instead, the whole TM segments were immersed in the forming oil lens, leading to the destruction of the ER-LD neck structure. In particular, the residues at the end of TM segments lost contact with the outer membrane during oil coalescence when a seipin-free oil lens approached the seipin ring and merged with the oil lens contained within it. The loss of the ER-LD neck structure resulted in a flat oil lens, as shown in our CG system that did not include seipin or that had seipin without TM segments (*Figure 4*). In contrast, the ER-LD neck structure was maintained during oil coalescence when *r* was large, as in the previous CG simulations. TG molecules in a seipin-free oil lens were transferred to the seipin in a controlled manner without disrupting the ER-LD neck structure. Collectively, the cell experiments and CG simulations suggest that the conserved, charged residues located at the borders of seipin TM segments maintain the ER-LD neck structure during oil coalescence and promote LD growth.

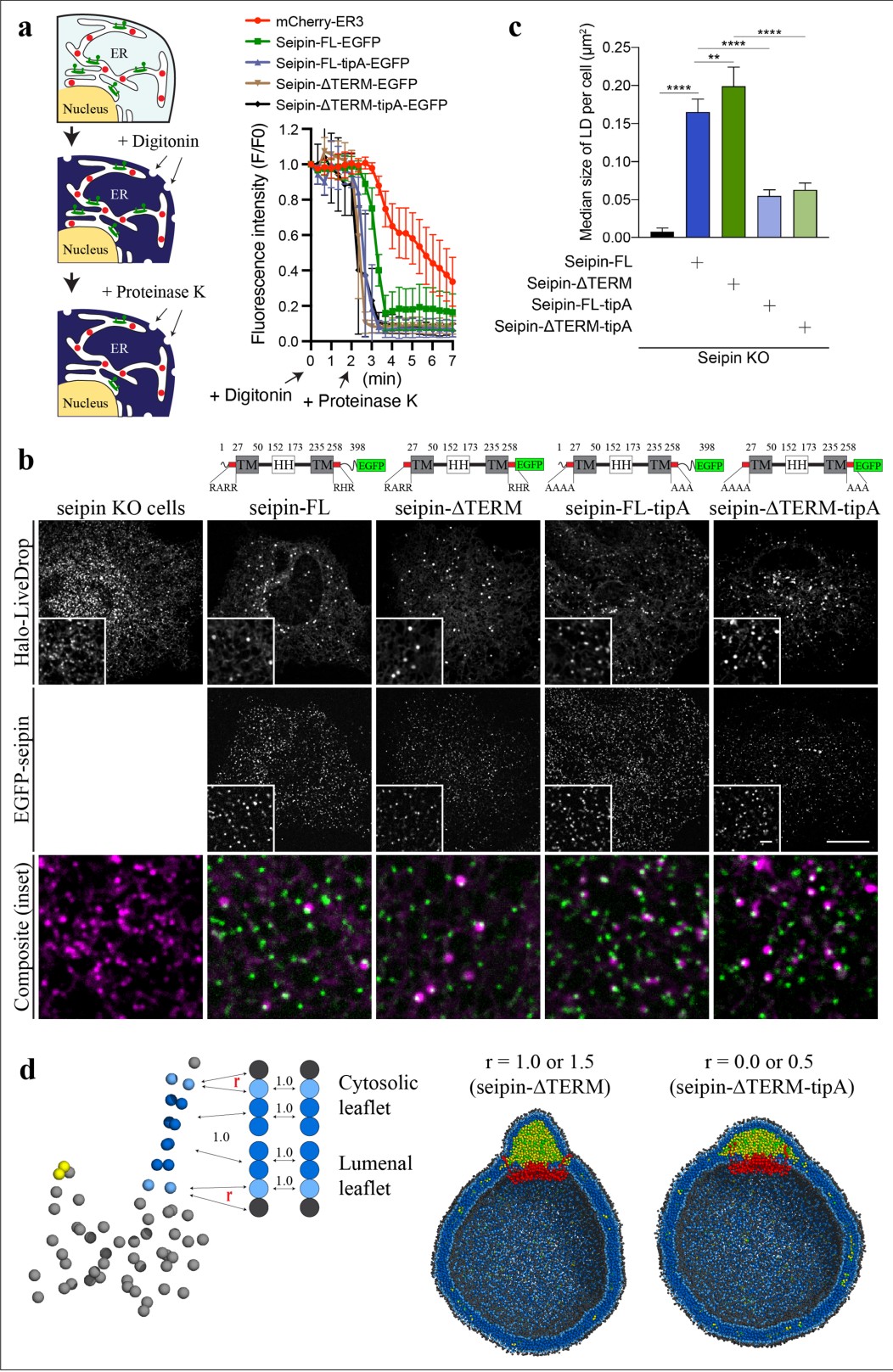

**Figure 6.** Nonconserved, cytosolic tails of human seipin are dispensable for the function, while the conserved, positively charged residues at the ends of seipin transmembrane (TM) segments are critical for lipid droplet (LD) maturation. (**a**) Schematic representation of a fluorescence protease protection (FPP) assay (left). Low concentration of digitonin allows permeabilization of the plasma membrane without disrupting the endoplasmic reticulum

*Figure 6 continued on next page*

*Figure 6 continued*

(ER) membrane. Application of proteinase K selectively cleaves cytosolically exposed fluorescent protein (green) without affecting lumenally exposed fluorescent protein (red). Quantification of the fluorescence intensities of the whole time series of the FPP assay (right). (mean ± standard error of the mean [SEM]). (**b**) Confocal imaging of live seipin knockout (KO) SUM159 cells transiently transfected with various seipin constructs fused with EGFP and Halo-LiveDrop (stained with JF549). The cells were preincubated with 0.5 mM oleic acid for 1 hr prior to image acquisition. Scale bars, full-size, 15 µm; inserts, 2 µm. (**c**) Quantification of size of LDs per cell shown in (b) $n = 4$ cells. More than 300 LDs were analyzed in each sample. Median with interquartile range. ****$p < 0.0001$, **$p < 0.01$ were calculated by unpaired *t*-test. (**d**) CG molecular dynamics (MD) simulations with various attraction scaling factors (*r*). The spherical bilayer has a diameter of 40 nm and contains 6% mol TG. The elastic network model (ENM) model of human seipin with a spring constant of 0.2 kcal/mol/Å$^2$ was used.

The online version of this article includes the following figure supplement(s) for figure 6:

**Figure supplement 1.** Sequence of human seipin.

## Discussion

Seipin is a critical protein that orchestrates LD formation (*Arlt et al., 2022*; *Chung et al., 2019*; *Klug et al., 2021*; *Prasanna et al., 2021*; *Salo et al., 2020*; *Salo et al., 2019*; *Wang et al., 2016*; *Zoni et al., 2021b*). Recent joint computational and experimental studies reported that the HH of seipin facilitates TG nucleation (*Klug et al., 2021*; *Prasanna et al., 2021*; *Zoni et al., 2021b*). However, little is known about the roles of seipin TM segments despite their experimentally confirmed importance for seipin's function (*Arlt et al., 2022*; *Chung et al., 2019*). Capitalizing on highly CG models of lipids and seipin, we investigate LD maturation, showing that a cage-like geometry of seipin TM segments forms a constricted neck, converting a planar oil lens into a unique ER-LD, facilitating LD growth. In contrast, the system that lacked seipin or contained seipin without TM segments resulted in a flat oil lens with high anisotropy.

Using cell-based experiments and CG simulations, we further identified certain essential and dispensable parts of human seipin. We show that the nonconserved N- and C-terminal cytosolic regions of human seipin are not required for LD formation. Therefore, truncated seipin models used in the current and previous computational studies are reasonable due to the dispensability of the cytosolic tails (*Klug et al., 2021*; *Prasanna et al., 2021*; *Zoni et al., 2021b*). We also provide evidence that the conserved, positively charged residues located at the borders of the seipin TM segments are crucial for LD maturation. Mutating those residues to alanine resulted in an intermediate LD phenotype with more and nonuniform shaped LDs than those formed in wild-type cells. However, these mutations did not alter seipin's membrane topology. In the CG simulations, seipin TM segments were immersed in an oil lens when interactions between those residues and PL interfacial and head groups became reduced.

Based on our data, we propose a model in which the positively charged residues, located at the borders of seipin TM segments, anchor the TM segments at the cytosolic side of the membrane (*Figure 7*). This was particularly important in maintaining the ER-LD neck structure during oil coalescence. When a seipin-free oil lens approaches a seipin-positioned oil lens, the TM segments should keep their positions in the bilayer to maintain the ER-LD neck structure. Strong electrostatic interactions between the seipin residues at the borders of seipin TM segments and the membrane interface of the cytosolic leaflet inhibited rapid coalescence. Instead, TG in the seipin-free lens was transported to the seipin-positioned oil lens in a controlled manner. A slow coalescence at the seipin site is implicated in anisotropy analysis in *Figure 4*. If the interactions between the seipin TM tip residues and the cytosolic leaflet are small, the seipin TM segments become immersed in the lens during swift coalescence, affecting LD maturation (*Figure 6d*).

We investigated protein–lipid interactions and lipid dynamics using the AA simulation. Hydrophobic interactions between seipin TM segments and HH with TG are the main driving force of TG nucleation, in conjunction with hydrophilic interactions between the TG glycerol moiety and protein residues. It is worth noting that different force fields consistently report a strong attraction between S166 and TG glycerol moieties (*Prasanna et al., 2021*; *Zoni et al., 2021b*). During LD growth, lipids or proteins such as *LiveDrop* migrate from the ER onto LD through the populated TM region (*Olarte et al., 2020*; *Wang et al., 2016*). The diameter of the ER-LD neck structure during the LD growth

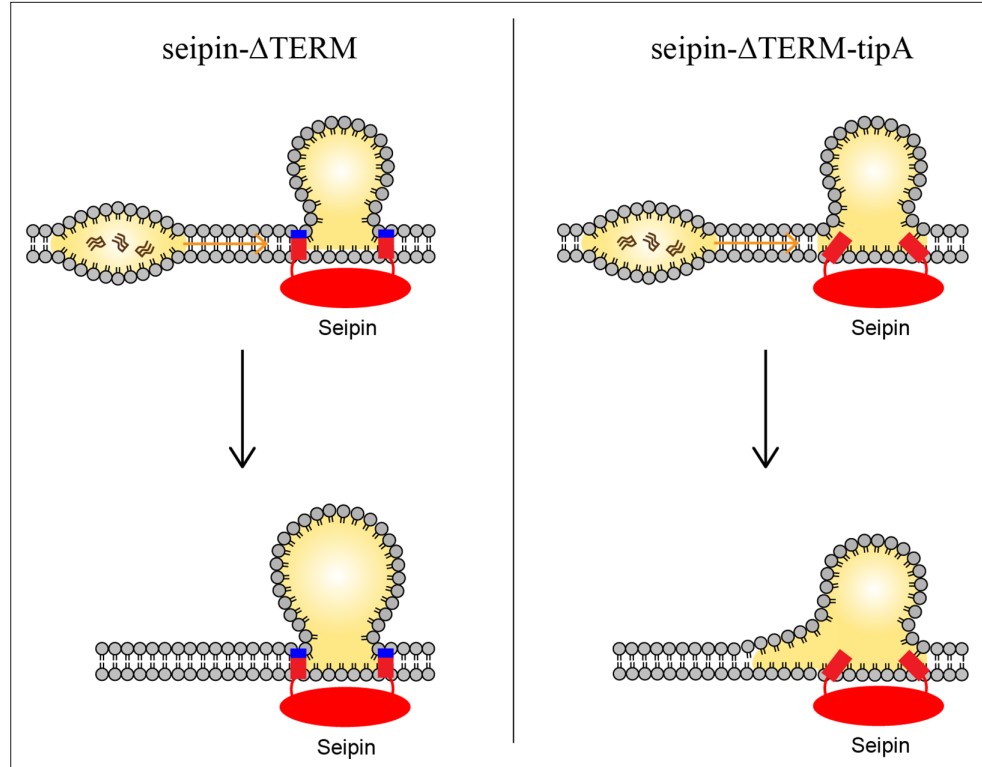

**Figure 7.** Model for the role of the charged residues at the end of seipin transmembrane (TM) segments during oil coalescence. The left side shows the seipin TM segments maintain the bilayer thickness around the endoplasmic reticulum (ER)-lipid droplet (LD) neck structure. The right side shows the TM segments are immersed in the oil lens during oil coalescence because positively charged residues at the end of seipin TM segments (blue) are mutated to alanine.

phase is larger than during the initial nucleation stage as demonstrated in the CG simulations. The widely spread TM segments will promote lipid and protein influx to LDs.

Finally, we discuss the limitations of our study. First, LDAF1 is not included in our simulations as its structure and number in the seipin oligomer are not identified yet. LDAF1 is predicted to have a double hairpin structure, which increases the density of TM segments inside the seipin oligomer. Therefore, including LDAF1 in simulations will likely change protein–lipid interactions in the system and lipid diffusion. Second, the CG simulations benefited from highly CG models to simulate large systems and study advanced LD biogenesis steps. However, seipin sequence-specific features are missing in the seipin model. Also, since this is a phenomenological model, the quantities calculated are not directly related to the underlying AA systems.

Taken together, we suggest a model in which seipin TM segments are key for LD biogenesis by assisting TG nucleation, controlling lipid diffusion, defining the boundary of the forming oil lens, maintaining the ER-LD neck structure, and controlling oil coalescence. Our study thus provides a broader and deeper understanding of the roles of the TM segments, critical for seipin function.

# Materials and methods
## AA MD simulation

The seipin simulation in a POPC bilayer including 6% triolein was carried out for 3 μs. The equilibrated bilayer structure was taken from the previous work (***Kim and Voth, 2021***). Seipin has a HH ring at the center with a radius of ~2 nm in the lumenal leaflet. We first placed 20 POPC molecules inside the seipin HH ring with their orientations consistent with other lumenal POPC molecules using PACKMOL (***Martínez et al., 2009***). Those PLs remained trapped inside the ring during the simulation (proteinized PLs). We put the seipin complex at the membrane center and removed any lipid molecules within

0.9 Å of seipin atoms. The equilibrium protocol provided by the CHARMM-GUI interface was used (*Lee et al., 2016*). Additionally, 100 ns of equilibration was carried out, restraining the positions of the backbone atoms of the lumenal domain (Val60–His219) and the $Z$ positions of phosphorus atoms with a spring constant of 20 kJ/mol/nm$^2$. The total numbers of POPC and TG molecules were 797 and 48, respectively.

The simulation was run by GROMACS 2020 (*Van Der Spoel et al., 2005*) with a Lennard–Jones (LJ) cutoff-free version of CHARMM36 (*Yu et al., 2021a*; *Yu et al., 2021b*). The modified TG parameters that reproduced the interfacial tension against water were used (*Kim and Voth, 2021*). Simulations were evolved with a 2-fs time step. The long-range electrostatic and LJ interactions were evaluated with the Particle Mesh Ewald algorithm, with the real-space cutoff distance of 1.0 nm (*Essmann et al., 1995*). Bond involving a hydrogen atom was constrained using the LINCS algorithm (*Hess, 2008*). A temperature of 310 K and a pressure of 1 bar were maintained with the Nose–Hoover thermostat and the Parrinello–Rahman barostat, respectively (*Hoover, 1985*; *Nosé, 1984*; *Parrinello and Rahman, 1981*). The coupling time constants of 1 and 5 ps were used, respectively. A compressibility of 4.5 (please add multiplication symbol) $10^{-5}$ bar$^{-1}$ was used for semi-isotropic pressure coupling.

## Coordination number analysis

To study protein–lipid interactions, we first reduced the resolution of the AA simulation by mapping each POPC molecule into 11 molecular groups and each TG molecule into 13 groups. In this mapping scheme, each POPC molecule had the choline head group, phosphate group, glycerol moiety and four tail groups for each acyl chain. Similarly, each TG molecule had the glycerol moiety and four tail groups for each acyl chain. Each protein residue was mapped into one backbone and one side chain group. For each amino acid, the coordination number between the side chain group and membrane groups of PL or TG was calculated by $s_M = \sum_M \sum_a \left[ 1 - (r_a/r_0)^6 \right] / \left[ 1 - (r_a/r_0)^{12} \right]$, where $M$ represents PL or TG and $a$ represents a CG atom belonged to $M$. The parameter $r_0$ was set to 0.4 nm and $r_a$ is the distance between the side chain group and membrane group (atom $a$). The normalized coordination number by molecule or the coordination number per molecule ($\| s \|$) was computed by diving the coordination number by the number of molecules of PL or TG. The normalized coordination number by CG atom or the coordination number per CG atom ($\| s_A \|$) was calculated by dividing the coordination number by the number of atoms of atom $A$.

## Diffusion coefficient

Using the AA trajectory, we carried out the diffusion coefficient calculation with the MDAnalysis library (*de Buyl, 2018*; *Michaud-Agrawal et al., 2011*). When calculating diffusion coefficients, we translated the system such that the center of the mass of the lumenal domain of seipin was at the origin in each frame. Therefore, the diffusion coefficient of PL or TG reported here represents the relative diffusion coefficient with respect to the center of mass of the protein. The trajectory was divided into three trajectories, each of which was 1 µs long. In each trajectory, PL or TG molecules were categorized into three classes, based on the average $XY$ distance from the origin. The first class of lipids was located within 3.5 nm from the origin, slightly greater than the radius of the HH ring. The second class of lipids was located between 3.5 and 7.0 nm from the origin, where they mainly interacted with the TM segments. Finally, the lipids that were further than 7.0 nm were considered lipids in the protein-free zone as they did not interact with the protein. The position-dependent diffusion coefficients were reported by calculating diffusion coefficients in each class. Three equal-length blocks were used to report the average and standard errors.

## MARTINI CG model simulation

MARTINI protein and lipids (v2.2) were used for MARTINI CG simulations (*de Jong et al., 2013*; *Marrink et al., 2003*; *Monticelli et al., 2008*), run by GROMACS 2018 (*Van Der Spoel et al., 2005*). Simulations were evolved with a 20-fs time step. A cutoff distance of 1.1 nm was used for electrostatic and LJ interactions. A temperature of 310 K and a pressure of 1 bar were maintained with the V-rescale thermostat and the Parrinello–Rahman barostat, respectively (*Bussi et al., 2007*; *Parrinello and Rahman, 1981*). The coupling time constants of 1 and 12 ps were used, respectively. A compressibility of 3.0 (please add multiplication symbol) $10^{-4}$ bar$^{-1}$ was used for semi-isotropic pressure coupling. A relative dielectric constant of 15 was used.

## CG model simulations

We used a previously developed CG model for PL and TG with each molecule consisting of four CG beads (*Grime and Madsen, 2019*; *Kim et al., 2022b*). An angle parameter of 0.5 $k_B T$ for PL was used in this study. A CG model for seipin was constructed by linearly mapping four amino acids into each CG bead. Such a resolution was chosen to match the resolution of the CG lipids, preventing hydrophobic mismatch. We also placed 24 PL molecules inside the HH ring with their orientations consistent with the other PL molecules in the lumenal leaflet. Three models were constructed with different elastic networks with bond potentials $k\left(r - r_0\right)^2$ where $k$ is the spring constant and $r_0$ is the equilibrium bond length. The first two models utilized a standard ENM with a spring constant of 0.2 or 2 kcal/mol Å$^2$ and with a distance cutoff of 15 Å. The third model utilized the hENM that more correctly represented the fluctuations of seipin in the underlying AA simulation (*Lyman et al., 2008*). From the AA MD simulation, we first made a concatenated, aligned seipin monomer trajectory and obtained the hENM with a cutoff distance of 12 Å. The hENM was applied to each monomer. A spring constant of 0.1 kcal/mol Å$^2$ was applied to the CG pairs that did not have hENM if the distance is less than 11 Å. To achieve the known stability of seipin in a bilayer membrane, nonbonded protein–lipid interactions were based on the lipid–lipid interactions (*Grime and Madsen, 2019*; *Kim et al., 2022b*). Protein atoms located in the hydrophobic phase interacted with the PL tail atoms with the equal attraction strength of the pair between PL tail and PL tail atoms or PL interfacial and PL interfacial atoms (scaling factor = 1). Protein atoms located at the membrane interface attracted PL interfacial atoms with a scaling factor of 1.5 unless otherwise noted. The central region of TM segments and two HH beads attracted TG atoms with a scaling factor of 1.5. Every CG bead carried a mass of 200 g/mol and no charge. Spherical bilayers with a diameter of 40 or 60 nm were constructed, containing 2% TG or 6% TG, followed by the placement of seipin and removal of lipids that had a close contact with seipin.

The CG simulations were run by LAMMPS MD software with tabulated CG potentials (*Plimpton, 1995*). Simulations were evolved with a 50-fs time step. Temperature was maintained at 310 K by the Langevin thermostat with a coupling constant of 100 ps (*Schneider and Stoll, 1978*). The cutoff distance of nonbonded interaction was 1.5 nm. The initial structures of CG simulations were prepared with the MDAnalysis library (*Michaud-Agrawal et al., 2011*).

## Nucleation percentage and anisotropy

We calculated the nucleation percentage and anisotropy as explained in *Kim et al., 2022b*. In short, the nucleation percentage was defined as the ratio of the number of TG molecules in the largest cluster to the number of TG molecules in the system. The distance cutoff of 2 nm was used for clustering TG molecules. The anisotropy was calculated by diagonalizing the moment of inertia tensor of the largest TG cluster. The anisotropy of 0 represents a spherical shape, and the anisotropy of 0.25 represents a planar shape.

## Cell culture

SUM159 breast cancer cells (RRID:CVCL_5423) were obtained from the laboratory of Dr. Tomas Kirchhausen (Harvard Medical School) and were maintained in DMEM/F-12 GlutaMAX (Life Technologies) supplemented with 5 µg/ml insulin (Cell Applications), 1 µg/ml hydrocortisone (Sigma), 5% FBS (Life Technologies 10082147; Thermo Fisher), 50 µg/ml streptomycin, and 50 U/ml penicillin. Cell lines were tested monthly for mycoplasma contamination using the PCR Mycoplasma Test Kit (ABM Cat# G238) and always came back negative. The cell lines were authenticated via STR profiling.

## Plasmid construction

For plasmid construction, all PCRs were performed using PfuUltra II Fusion HotStart DNA polymerase (#600672, Agilent Technologies) and restriction enzymes were from New England Biolabs. The fragment DNA of tipA mutant construct was synthesized (gBlock, Integrated DNA Technologies).

## Fluorescence microscopy

Cells were plated on 35-mm glass-bottom dishes (MatTek Corp). Imaging was carried out at 37°C approximately 24 hr after transection. Before imaging, cells were transferred to prewarmed Fluoro-Brite DMEM supplemented with 2 mM GlutaMAX (#35050061, Thermo Fisher Scientific), 5 mg/ml insulin (Cell Applications), 1 mg/ml hydrocortisone (Sigma Aldrich), 5% fetal bovine serum (Life Technologies 10082147; Thermo Fisher), 50 mg/ml streptomycin, and 50 U/ml penicillin.

Spinning-disk confocal microscopy was performed using a Nikon Eclipse Ti inverted microscope equipped with Perfect Focus, a CSU-X1 spinning-disk confocal head (Yokogawa), Zyla 4.2 Plus scientific complementary metal–oxide semiconductor cameras (Andor, Belfast, UK), and controlled by NIS-Elements software (Nikon). To maintain 85% humidity, 37°C and 5% $CO_2$ levels, a stage top chamber was used (Okolab). Images were acquired through a 603 Plan Apo 1.40 NA objective or 1,003 Plan Apo 1.40 NA objective (Nikon). Image pixel size was 0.065 mm. Green or red fluorescence were excited by 488 or 560 nm (solid state; Andor, Andor, Cobolt, Coherent, respectively) lasers. All laser lines shared a quad-pass dichroic beamsplitter (Di01-T405/488/568/647, Semrock). Green and red emission was selected with FF03-525/50 or FF01-607/36 filters (Semrock), respectively, mounted in an external filter wheel. Multicolor images were acquired sequentially.

## Fluorescence protease protection assay

The membrane topology of seipin constructs were determined by fluorescence protease protection assay as described in *Lorenz et al., 2006*. Briefly, cells were washed three times with 2 ml of KHM buffer (110 mM potassium acetate, 20 mM HEPES [4-(2-hydroxyethyl)-1-piperazineethanesulfonic acid] (HEPES), pH 7.2, 2 mM $MgCl_2$) at room temperature. 40 µM digitonin and 100 µg/ml Proteinase K were added to the media during imaging acquisition to lyse plasma membrane and kill fluorescence proteins, respectively.

## Imaging quantification

LD size quantification was done with the fiji software. Individual images were first converted to a binary mask using 'Threshold' plugin in fiji with Otsu method. Then, LD sizes were measured by the fiji plugin 'Particle Analysis'.

## Protein expression and purification

The complex of LDAF1-FLAG and seipin (1–310) were expressed in suspension cultures of Expi293F cells (Life Technologies) which were cultured in Expi293 Expression Medium (#A1435102, Gibco) at 37°C under 8% $CO_2$ and 80% humidity in Multitron-Pro shaker at 125 rpm. When cell density reached $2.5 \times 10^6$ cells per ml, the pCAG-LNK plasmids were transfected into the cells. For 2 l of cell culture, 2 mg of plasmids were premixed with 6 mg of 25 kDa linear polyethyleneimine (Polysciences) in 200 ml of Opti-MEM medium for 30 min at room temperature before transfection. At 16 hr after transfection, 10 mM sodium butyrate were added to boost protein expression. To enrich a status of the LDAF1–seipin complex that does not contain neutral lipids inside the complex, cells were treated with 50 µ DAGT1 and DGAT2 inhibitors for 48 hr and with 5 µM Triacsin C for 12 hr prior to cell harvest. At 48 hr after transfection, cells were collected, and cell pellets were snap frozen by liquid nitrogen and stored at −80°C.

All purification procedures were performed at 4°C. Cell pellet was thawed and resuspended in the Buffer A (50 mM Tris–HCl pH 8.0, 150 mM NaCl, 5 mM $MgCl_2$, 10% vol/vol glycerol) supplemented with cOmplete Protease Inhibitor Cocktail tablet, ethylenediaminetetraacetic acid (EDTA)-Free (Roche). Cells were lysed by sonication. The cell debris were removed by centrifugation at 5000 $\times$ g for 15 min. To get membrane fractions, the supernatant was subjected to centrifugation in a Ti45 rotor (Beckman) at 43,000 rpm for 1 hr. The membrane pellet was collected and homogenized with a Dounce homogenizer in Buffer A supplemented with 1% Lauryl Maltose Neopentyl Glycol (LMNG), and cOmplete Protease Inhibitor Cocktail tablet, EDTA-Free, and membranes were solubilized with gentle rocking for 1.5 hr. Insoluble material was then removed by centrifugation at 43,000 rpm for 35 min. The supernatant was incubated with 1.2 ml of anti-FLAG M2 resin (Sigma) for 1.5 hr. The resins were then collected and washed with 12 ml of Buffer A with 0.05% digitonin and the proteins were eluted with 3 ml of washing buffer containing 0.2 mg/ml of 3xFLAG peptide (Sigma). The eluted protein was concentrated and PMAL C8 was added at a mass ratio of 3–1 protein. Detergent was

removed by biobeads SM-2 overnight. The sample was further purified by size-exclusion chromatography on a Superose 6 3.2/300 Increase column, equilibrated with buffer containing 25 mM HEPES, pH 7.4, 150 mM NaCl. Peak fractions were pooled and concentrated to 2 mg/ml for cryo-EM analysis.

## Electron microscopy sample preparation and data acquisition

For cryo-EM analysis, the concentrated sample was incubated with MS(PEG)12 methyl-PEG-NHS-ester (Thermo Fisher) at a 1:10 molar ratio for 2 hr on ice to reduce aggregation of particles on the grids. PEGylated sample (3 µl) was applied to a glow-discharged quantifoil grid (1.2/1.3, 400 mesh). The grids were blotted for 2.5 s at ~90% humidity and plunge-frozen in liquid ethane using a Cryo-plunge 3 System (Gatan).

Cryo-EM data were collected on a Krios operated at 300 kV and equipped with a K3 Summit direct electron detector (Gatan) at Harvard Medical School. All cryo-EM movies were recorded in counting mode using SerialEM. The nominal magnification of ×81,000 corresponds to a calibrated physical pixel size of 1.06 Å. The dose rate was 22.8 electrons/Å2 s. The total exposure time was 2.2 s, resulting a total dose of 50.6 electrons/$Å^2$ fractionated into 49 frames. The defocus range for the sample was between 0.8 and 2.5 µm.

## Image processing

Dose-fractionated movies were subjected to motion correction, using the program MotionCor2 (*Zheng et al., 2017*). A sum of all frames of each image stack (49 total) was calculated by following a dose-weighting scheme and used for all image-processing steps except for defocus determination. The program Gctf (*Zhang, 2016*) was used to estimate defocus values of the summed images from all movie frames without dose weighting. Particles were autopicked by Gautomatch (http://www.mrc-lmb.cam.ac.uk/kzhang/). After manual inspection and sorting to discard poor images, classifications were done in Relion 3.0 (*Zivanov et al., 2018*). Particles were extracted and subjected to one round of reference-free 2D classification to remove false picks and obvious junk classes. The resulting particles were subjected to one round of global 3D classification without symmetry applied. Only one class with nice features for the luminal domains was selected for modeling. This class also showed some TM density for 1–2 copies of seipin.

## Seipin structure

We modeled the seipin structure (Arg23–Arg265) based on previously determined structure of lumenal domain and our cryo-electron microscopy data (*Figure 1—figure supplement 1*). The seipin structure contained the lumenal domain (Val60–His219), which was previously resolved (*Chung et al., 2019*; *Yan et al., 2018*), and the partially resolved TM segments. However, due to the low resolution of the TM region, we were not able to identify residues in the TM segments. Instead, we used the rough locations of the TM helices in our modeling. We assumed the residues that corresponded to the N- and C-terminal TM helical structures were Leu29–Gly50 and Ala235–Val258, respectively. The missing residues from Ser51 to Val60 were determined with Modeller (*Fiser et al., 2000*). The structure from Phe220 to Phe230, referred to as a switch region, was homology modeled using a yeast seipin structure as a reference because this region is highly conserved and predicted to be folded similarly (*Arlt et al., 2022*). The structure of the switch region was helical, and its helicity was further supported by the PSIPRED (*McGuffin et al., 2000*), TMHMM (*Krogh et al., 2001*), TMpred, and Phyre2 (*Kelley et al., 2015*) servers. The missing residues, Pro231–Cys234, were modeled with Modeller (*Fiser et al., 2000*). The human seipin model used in this study is available at https://github.com/ksy141/seipin (copy archived at swh:1:rev:795caa3e7a96359a4c4d27547272dd80d921568e; *Kim, 2022a*).

## Acknowledgements

This research was supported by National Institutes of Health (NIH) grants R01-GM063796 (to GAV), NIH R01-GM124348 (to RVF), and NIH R01-GM097194 (to TCW). The computer simulations were performed on the Stampede2 supercomputer at the Texas Advanced Computing Center and the Bridges2 supercomputer at the Pittsburgh Supercomputing Center (PSC) through allocation MCA94P017 with resources provided by the Extreme Science and Engineering Discovery Environment (XSEDE) supported by NSF grant ACI-1548562. We also utilized computational resources on the Midway3 supercomputer at the University of Chicago. JC is a fellow of the Damon Runyon Cancer

Research Foundation. TCW is a Howard Hughes Medical Institute Investigator. We thank Xudong Wu for performing cryo-electron microscopy imaging and critical discussion. SK acknowledges Chenghan Li and Sriramvignesh Mani for critical discussion.

## Additional information

### Funding

| Funder | Grant reference number | Author |
|--------|------------------------|--------|
| National Institutes of Health | | Robert V Farese<br>Tobias C Walther<br>Gregory A Voth |

The funders had no role in study design, data collection, and interpretation, or the decision to submit the work for publication.

### Author contributions

Siyoung Kim, Conceptualization, Formal analysis, Investigation, Writing - original draft, Writing - review and editing; Jeeyun Chung, Formal analysis, Investigation, Writing - original draft, Writing - review and editing; Henning Arlt, Alexander J Pak, Resources, Writing - review and editing; Robert V Farese, Tobias C Walther, Funding acquisition, Investigation, Supervision, Writing - review and editing; Gregory A Voth, Conceptualization, Funding acquisition, Investigation, Supervision, Writing - review and editing

### Author ORCIDs

Siyoung Kim http://orcid.org/0000-0003-4670-9747
Jeeyun Chung http://orcid.org/0000-0002-4617-7181
Robert V Farese Jnr, http://orcid.org/0000-0001-8103-2239
Gregory A Voth http://orcid.org/0000-0002-3267-6748

### Decision letter and Author response

Decision letter https://doi.org/10.7554/eLife.75808.sa1
Author response https://doi.org/10.7554/eLife.75808.sa2

## Additional files

### Supplementary files

- Transparent reporting form

### Data availability

Numerical data represented as a graph in this manuscript are available at https://github.com/ksy141/seipin.

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
