## [Editor Report]

Kim et al., investigate interactions between Seipin transmembrane domains and triacylglycerol using molecular dynamics simulations. They identify the leading steps in droplet formation and provide a physical basis for understanding the initial phases of this process, highlighting the importance of transmembrane helices in the function of seipin protein. This paper will be of interest to cell biologists and biophysicists aiming to unveil and understand how lipid droplets are formed inside cells. The topic is important given that lipid droplets are key organelles used for energy storage, and that the failure in their formation can result in various metabolic diseases.

---

## [Decision Letter]

**Decision letter after peer review:**

Thank you for submitting your article “Seipin transmembrane segments critically function in triglyceride nucleation and lipid droplet budding from the membrane” for consideration by *eLife*. Your article has been reviewed by 3 peer reviewers, and the evaluation has been overseen by a Lucie Delemotte as Reviewing Editor and José Faraldo-Gómez as Senior Editor. All reviewers have opted to remain anonymous.

The reviewers have discussed their reviews with one another, and the Reviewing Editor has drafted this letter to help you prepare a revised submission. We appreciate that the revisions required might entail a substantial effort, which might also exceed the typical scope and timeframe permitted for revisions at *eLife*. However, editors and reviewers recognize the interest and originality of the work; therefore, rather than declining this submission, we would like to invite you to revise the manuscript in a manner that convincingly resolves the critical issues raised by the reviewers. To enable you to do so, we will reconsider your revised manuscript irrespective of the timeframe required for these improvements. Please also note it is probable that your revised manuscript will be sent to the same reviewers for a re-evaluation.

Essential revisions:

1) A more thorough investigation of the quality of the models of the transmembrane regions of Seipin.

2) Improved sampling of the conformational space in atomistic simulations (introducing replicas).

3) A thorough characterization of the role of the charged residues and the residues in transmembrane position on protein expression, membrane insertion and topology.

4) A list of concrete predictions from the coarse-grained model that can be tested experimentally or by atomistic MD simulations.

5) Release of data and models to be openly available to the community.

*Reviewer #1 (Recommendations for the authors):*

Whilst a well-structured project, and representing a considerable amount of data, I have a couple of concerns on how well the experiments support the findings.

Chiefly, I am uncertain of how confident we can be of the modelled TM regions. Whilst fine to include, I feel that the bulk of the figures which interpret these regions are at risk of over-interpreting what might be low-accuracy input models. Seeing as even a small change in TM helix angle or register can cause dramatic artefacts, the model here seems potentially insufficient for use.

At the very least the following points should be addressed:

– These models should be made available for download, for verification and use by other researchers.

– The packing of side chains in the TM domains should be shown, as the image in Figure 1a has very limited use.

– How do the helices compare to the recent structure of the seipin from yeast [Klug et al., 2021, Nat Comms]? Are there any conserved geometries or side chain packing that could be used to refine or support the model used here?

– It is stated that the structural density in S1 was used to guide the modelling, but from the figure it is impossible to see how this was done. If there is reasonable density for the TM helices that can be shown, it should be. Otherwise, this comparison is misleading.

– The principal component analysis in Figure 3 suggests huge movements of the TM helices. This might be due to naturally high dynamics, however might also reflect a low quality input model. Ideally, some other data could corroborate these dynamics, but failing that at the very least the progression of each pair of TM helices (out of 11) should individually be projected on PC1 and PC2. This would show if the dramatic dynamics seen in the figure is happening in each subunit (which would support the author's claims) or if each subunit is doing something different. This would both show that the simulation is not ergodic, and also suggest that subunits are "collapsing" into different conformations, so possibly a result of an issue in the input model.

The second part, which focuses on the lens formation, is a neat piece of work, and directly addresses the biological question of LD formation. The outcomes from this section seem plausible, however I have a couple of points which should be addressed:

– The simulations are very coarse-grained, and it's open for debate how well a molecule with 3 tails like TAG can be represented by a 4-bead model. Ideally, some "bridging" simulations using something like Martini could be used on a smaller system to demonstrate that the same observations can be made.

– Are the dynamics of the CG system the same as the AT system, as in, are the same extremes in TM angles observed?

Combining these last two points – I wonder if random modelling of the TMs would not produce the same results, as the effect seems to be mostly due to the length of the TM helices and the position of the basic residues. If so, I'd argue that the first part of the study is not really necessary for the study, and overall makes the work weaker.

*Reviewer #2 (Recommendations for the authors):*

– The conclusions of the study rely heavily on a structural model that is described only superficially. How was the density map/model presented in Figure S1 determined? The transmembrane residues were modelled based on the structure of the yeast protein that is not publicly available (Arlt et al., BioRxiv 2021). How does this model compare with the predicted Alphafold structure? Information on the transmembrane segment residues and the transmembrane segment conformational changes would be particularly relevant.

– It is stated that the switch region leads to lower lipid permeability on the luminal leaflet. This is potentially interesting but could the presence of the bulky luminal domain explain this behaviour (and not the Switch region per se)? Please comment.

– It would be interesting to show the analysis on the evolutionary conservation of the transmembrane residues. Additional mutagenesis of conserved residues to demonstrate their functional relevance would also be helpful.

– In Figure 6 the conclusion is "Therefore, high protein density at the seipin site provides for collective and cooperative interactions with TG, catalyzing TG nucleation". This appears unjustified since Seipin monomers were not used as control. Instead, it seems that a single TM protein lacking the TAG-binding hydrophobic helix was used.

– The identification of the charged residues flanking transmembrane segments is potentially interesting. However, given the role of charged residues in protein topogenesis the authors should exclude that the phenotype is not indirect due to poor expression or changes in membrane topology.

– According to Yan et al., (2019), Seipin luminal domain (without the TMs) expresses well and appears to oligomerize. This contrasts with the what is described on page 14: “Finally, we simulated the seipin model that only contained the luminal domain. Such a complex does not form a mobile focus in cells likely because it fails to form an oligomer, it is not stable in bilayers, or it is degraded”. Please comment on the discrepancy.

*Reviewer #3 (Recommendations for the authors):*

The authors have done a very nice piece of work. Overall, the quality is high, the results are very interesting, and the discussion of the results is fresh and insightful.

Nonetheless, please let me propose a few ideas to improve the quality of the manuscript further.

The main weakness of the study is the limited sampling of the atomistic simulation model. If the reviewer is not mistaken, the authors did only one simulation. While its length was 3 microseconds, it is short given that the formation of a triglyceride (TG)-rich lipid phase is limited by lipid diffusion, which is a slow process, and since the transmembrane helices of seipin block diffusion, hence slowing down the formation of the TG-rich phase. This concern is increased by the fact that the development of the coarse-grained (CG) model is partly based on atomistic simulation data. The authors are encouraged to improve the sampling of atomistic simulations by carrying out a couple of additional simulations that would start from a different initial state / lipid configuration.

In Methods, the methodology of cryo-EM imaging spectroscopy is not presented. These descriptions would be useful, since the structure of the transmembrane region in the simulation models is based on cryo-EM imaging data (Figure S1) and educated guesses.

The results in Figure 4 are interesting since they present (first) evidence of the rate of lipid diffusion inside a cage-like protein complex. The lipid diffusion data outside but in the vicinity of the seipin oligomer, in turn, is consistent with previous studies (such as, Javanainen et al., (J Phys Chem Lett 8, 4308 (2017)(supplementary) and Niemela et al., (JACS 132, 7574 (2010)))), which could be briefly discussed here.

Page 13 ("To study the impact of the cage-like structure of the seipin oligomer and their TM segments in LD biogenesis…"). These simulations were carried out at 6 mol% TG, which is above the critical concentration of TG clustering. What happens if the TG concentration is 2 mol%, as in previous simulations discussed earlier in the paper? These additional simulations would be relevant, since they would clarify the relative importance of different seipin units in the clustering of TG (seipin luminal domain, different numbers of transmembrane helices).

Main conclusions of the paper are based on the CG model, and the conclusions are consistent with current experimental knowledge. However, since the CG model is phenomenological and its properties can be tuned (scaling factor, among others), the question is, what new insight the present CG simulation data can generate that could be tested in experiments and more detailed atomistic simulations? In other words, are the authors able to present experiments to test the key predictions in this paper?

---

## [Author Response]

Reviewer #1 (Recommendations for the authors):Whilst a well-structured project, and representing a considerable amount of data, I have a couple of concerns on how well the experiments support the findings.Chiefly, I am uncertain of how confident we can be of the modelled TM regions. Whilst fine to include, I feel that the bulk of the figures which interpret these regions are at risk of over-interpreting what might be low-accuracy input models. Seeing as even a small change in TM helix angle or register can cause dramatic artefacts, the model here seems potentially insufficient for use.

We appreciate the reviewer’s input. We understand the reviewer’s concern about the low-resolution of the TM segments. Although the cryo-EM is low-resolution, it provided valuable information such as the orientations and locations of the TM segments of seipin. Therefore, this information based on cryo-EM builds a higher quality of the seipin model than the previous papers, PNAS 2021, 118, 10 and PLoS biology 2021, 19, 1, e3000998, where they used a random guess. In our updated manuscript, reflecting the reviewer’s suggestion, we discussed this in more detail and removed the results and discussion of the TM movements.

At the very least the following points should be addressed:– These models should be made available for download, for verification and use by other researchers.

We have made the model publicly available.

– The packing of side chains in the TM domains should be shown, as the image in Figure 1a has very limited use.

We have now included a detailed structural model in Figure 1.

– How do the helices compare to the recent structure of the seipin from yeast [Klug et al., 2021, Nat Comms]? Are there any conserved geometries or side chain packing that could be used to refine or support the model used here?

We have compared our model against the recently resolved yeast structure in Figure 1.

– It is stated that the structural density in S1 was used to guide the modelling, but from the figure it is impossible to see how this was done. If there is reasonable density for the TM helices that can be shown, it should be. Otherwise, this comparison is misleading.

The TM helices are shown in Figure 1 —figure supplement 1, and the orientations of those helices were used in modelling our structure.

– The principal component analysis in Figure 3 suggests huge movements of the TM helices. This might be due to naturally high dynamics, however might also reflect a low quality input model. Ideally, some other data could corroborate these dynamics, but failing that at the very least the progression of each pair of TM helices (out of 11) should individually be projected on PC1 and PC2. This would show if the dramatic dynamics seen in the figure is happening in each subunit (which would support the author’s claims) or if each subunit is doing something different. This would both show that the simulation is not ergodic, and also suggest that subunits are “collapsing” into different conformations, so possibly a result of an issue in the input model.

The depicted PC1 and PC2 motions were associated with their maximum and minimum values; Therefore, they can look exaggerated. We would not expect all-atom simulations can show complete ergodicity of the TM segment movements even if the input model was very accurately resolved due to the limited time scale and the large size of seipin.

The second part, which focuses on the lens formation, is a neat piece of work, and directly addresses the biological question of LD formation. The outcomes from this section seem plausible, however I have a couple of points which should be addressed:– The simulations are very coarse-grained, and it’s open for debate how well a molecule with 3 tails like TAG can be represented by a 4-bead model. Ideally, some “bridging” simulations using something like Martini could be used on a smaller system to demonstrate that the same observations can be made.

We have now included additional higher CG resolution MARTINI simulations in Figure 3e; The 4-bead CG models are now published in J. Phys. Chem. B 2022, 126, 2, 453–462.

– Are the dynamics of the CG system the same as the AT system, as in, are the same extremes in TM angles observed?Combining these last two points – I wonder if random modelling of the TMs would not produce the same results, as the effect seems to be mostly due to the length of the TM helices and the position of the basic residues. If so, I’d argue that the first part of the study is not really necessary for the study, and overall makes the work weaker.

The RMSF from the CG simulations agrees reasonably well with that from the AA simulations; Also, the radius of the TM segments in CG simulations becomes larger due to TG nucleation and LD maturation, which was not possible to see in the more limited AA simulation. We feel the first part of the study is valuable and adds to the paper.

Reviewer #2 (Recommendations for the authors):– The conclusions of the study rely heavily on a structural model that is described only superficially. How was the density map/model presented in Figure S1 determined? The transmembrane residues were modelled based on the structure of the yeast protein that is not publicly available (Arlt et al., BioRxiv 2021). How does this model compare with the predicted Alphafold structure? Information on the transmembrane segment residues and the transmembrane segment conformational changes would be particularly relevant.

We have updated our methods section to explain how the density map/model was obtained in detail. Arlt et al., is now published in Nat Struct Mol Biol 2022. The Alphafold-predicted TM structure of human seipin seems to deviate from the expectations in the field. Despite its publicity, Alphafold is not a panacea, especially for membrane proteins. We have publicly made our model available and updated Author response image 1 to show the detailed structural model.

**Author response image 1. sa2fig1:** Structural comparison between AlphaFold2 (red) and our seipin model (green).

– It is stated that the switch region leads to lower lipid permeability on the luminal leaflet. This is potentially interesting but could the presence of the bulky luminal domain explain this behaviour (and not the Switch region per se)? Please comment.

We agree with the reviewer. We have updated our manuscript.

– It would be interesting to show the analysis on the evolutionary conservation of the transmembrane residues. Additional mutagenesis of conserved residues to demonstrate their functional relevance would also be helpful.

We feel this is interesting but beyond the current scope of the manuscript.

– In Figure 6 the conclusion is "Therefore, high protein density at the seipin site provides for collective and cooperative interactions with TG, catalyzing TG nucleation". This appears unjustified since Seipin monomers were not used as control. Instead, it seems that a single TM protein lacking the TAG-binding hydrophobic helix was used.

We performed a new simulation that includes only one seipin monomer. TG did not nucleate at 2 mol %. We have updated Figure 3d.

– The identification of the charged residues flanking transmembrane segments is potentially interesting. However, given the role of charged residues in protein topogenesis the authors should exclude that the phenotype is not indirect due to poor expression or changes in membrane topology.

We have included additional experiments to address this. See Figure 6a.

– According to Yan et al., (2019), Seipin luminal domain (without the TMs) expresses well and appears to oligomerize. This contrasts with the what is described on page 14: "Finally, we simulated the seipin model that only contained the luminal domain. Such a complex does not form a mobile focus in cells likely because it fails to form an oligomer, it is not stable in bilayers, or it is degraded". Please comment on the discrepancy.

It was hard to find an article only with the last name (Yan) of the first author and the publication year (2019). Given the context, we assumed the reviewer tried to refer to Dev. Cell. 2018, 47, 2, 248-256. We have carefully reviewed this paper again. However, this article does not include any in-vivo experiments with the seipin luminal domain (without the TMs).

Reviewer #3 (Recommendations for the authors):The authors have done a very nice piece of work. Overall, the quality is high, the results are very interesting, and the discussion of the results is fresh and insightful.Nonetheless, please let me propose a few ideas to improve the quality of the manuscript further.The main weakness of the study is the limited sampling of the atomistic simulation model. If the reviewer is not mistaken, the authors did only one simulation. While its length was 3 microseconds, it is short given that the formation of a triglyceride (TG)-rich lipid phase is limited by lipid diffusion, which is a slow process, and since the transmembrane helices of seipin block diffusion, hence slowing down the formation of the TG-rich phase. This concern is increased by the fact that the development of the coarse-grained (CG) model is partly based on atomistic simulation data. The authors are encouraged to improve the sampling of atomistic simulations by carrying out a couple of additional simulations that would start from a different initial state / lipid configuration.

We appreciate the reviewer’s positive feedback. We agree with the reviewer that the sampling of atomistic simulations was limited as this is a fairly large system and TG does not nucleate within the all-atom time scale. To address the reviewer’s concerns, we have argued the followings:

First, many lipids moved more than a dimension of the periodic box.

Second, this is the longest all-atom seipin simulation to date as a single trajectory.

Third, we have complemented our all-atom simulations with the MARTINI CG model simulations, which are now included in Figure 3e.

Finally, we have extended our all-atom simulations by 500 ns, and the results were consistent with the previous 3 microsecond simulation.

In Methods, the methodology of cryo-EM imaging spectroscopy is not presented. These descriptions would be useful, since the structure of the transmembrane region in the simulation models is based on cryo-EM imaging data (Figure S1) and educated guesses.

We have updated our Methods section.

The results in Figure 4 are interesting since they present (first) evidence of the rate of lipid diffusion inside a cage-like protein complex. The lipid diffusion data outside but in the vicinity of the seipin oligomer, in turn, is consistent with previous studies (such as, Javanainen et al., (J Phys Chem Lett 8, 4308 (2017)(supplementary) and Niemela et al., (JACS 132, 7574 (2010)))), which could be briefly discussed here.

We appreciate the reviewer’s input on this. We have discussed those aspects in our updated manuscript.

Page 13 ("To study the impact of the cage-like structure of the seipin oligomer and their TM segments in LD biogenesis…"). These simulations were carried out at 6 mol% TG, which is above the critical concentration of TG clustering. What happens if the TG concentration is 2 mol%, as in previous simulations discussed earlier in the paper? These additional simulations would be relevant, since they would clarify the relative importance of different seipin units in the clustering of TG (seipin luminal domain, different numbers of transmembrane helices).Main conclusions of the paper are based on the CG model, and the conclusions are consistent with current experimental knowledge. However, since the CG model is phenomenological and its properties can be tuned (scaling factor, among others), the question is, what new insight the present CG simulation data can generate that could be tested in experiments and more detailed atomistic simulations? In other words, are the authors able to present experiments to test the key predictions in this paper?

Although we did not indicate this in the manuscript, the idea of the mutant construct (seipin-∆TERM-tipA) was actually proposed to the experimentalists from the simulators based on the coarse-grained simulations. While we analyzed our coarse-grained simulations, we found that the charged residues at the borders seem to be important, and therefore we proposed the mutant construct (tipA construct) to our experimental collaborators. Consistent with our initial expectations, the mutant construct could not produce mature lipid droplets properly. To support this finding more, we added additional experiments during the revisions (Figure 6a).